# RNA Viruses in Aquatic Ecosystems through the Lens of Ecological Genomics and Transcriptomics

**DOI:** 10.3390/v14040702

**Published:** 2022-03-28

**Authors:** Sandra Kolundžija, Dong-Qiang Cheng, Federico M. Lauro

**Affiliations:** 1Asian School of the Environment, Nanyang Technological University, 50 Nanyang Avenue, Singapore 639798, Singapore; sandra001@e.ntu.edu.sg; 2Singapore Centre for Environmental Life Sciences Engineering, Nanyang Technological University, 60 Nanyang Drive, Singapore 637551, Singapore; dongqiang.cheng@ntu.edu.sg

**Keywords:** marine RNA virus, viral metatranscriptomics, metaviromics, dsRNA sequencing, viral ecology, viral diversity

## Abstract

Massive amounts of data from nucleic acid sequencing have changed our perspective about diversity and dynamics of marine viral communities. Here, we summarize recent metatranscriptomic and metaviromic studies targeting predominantly RNA viral communities. The analysis of RNA viromes reaffirms the abundance of lytic (+) ssRNA viruses of the order Picornavirales, but also reveals other (+) ssRNA viruses, including RNA bacteriophages, as important constituents of extracellular RNA viral communities. Sequencing of dsRNA suggests unknown diversity of dsRNA viruses. Environmental metatranscriptomes capture the dynamics of ssDNA, dsDNA, ssRNA, and dsRNA viruses simultaneously, unravelling the full complexity of viral dynamics in the marine environment. RNA viruses are prevalent in large size fractions of environmental metatranscriptomes, actively infect marine unicellular eukaryotes larger than 3 µm, and can outnumber bacteriophages during phytoplankton blooms. DNA and RNA viruses change abundance on hourly timescales, implying viral control on a daily temporal basis. Metatranscriptomes of cultured protists host a diverse community of ssRNA and dsRNA viruses, often with multipartite genomes and possibly persistent intracellular lifestyles. We posit that RNA viral communities might be more diverse and complex than formerly anticipated and that the influence they exert on community composition and global carbon flows in aquatic ecosystems may be underestimated.

## 1. Introduction

Viruses are vital parts of the marine ecosystem. They are major contributors to processes such as remineralization of carbon and nutrients and continuously regulate the diversity and abundance of complex microbial communities in the ocean, consisting of bacteria, archaea, and unicellular eukaryotes, protists [1,2,3,4]. Plankton protist communities largely consist of unicellular eukaryotic algae (also known as eukaryotic phytoplankton), heterotrophic flagellates and amoebas [5]. Viruses themselves are incredibly complex when it comes to genome type, replication, infection mode, morphology, and host preference. They infect cellular organisms in all three domains of life, as well as other viruses, and a single species can be infected by multiple viruses. In aquatic environments, eukaryotic phytoplankton is known to host very diverse viral communities [6,7]. Among aquatic bacteriophages, double-stranded (ds) DNA viruses are the most extensively studied and were thought to dominate prokaryotic virosphere [8]. Reports of single-stranded (ss) RNA and single-stranded (ss) DNA bacteriophages in environmental metaviromic and metatranscriptomic datasets [9,10,11,12,13] challenged this idea and suggested that bacteriophages have equally diverse genome architecture as eukaryotic viruses. On the basis of their genome architecture and replication strategy, viruses were classified in seven groups, now widely known as the Baltimore classes, a classification still used in parallel with the official viral taxonomy [14,15]. However, the molecular taxonomy has shown that these groups are not monophyletic, and for RNA viruses, the transition between single-stranded and double-stranded genome types has occurred multiple times during evolutionary history [16].

All RNA viruses are a part of the newly established viral realm, *Riboviria*. Within this realm, kingdom *Pararnavirinae* consists of RNA viruses with the reverse transcriptase (RT) and kingdom *Orthornavirinae* contains RNA viruses with the RNA-dependent RNA polymerase (RdRp). Kingdom *Orthornavirinae* splits in into five phyla on the basis of the phylogenetic analysis of RNA-dependent RNA polymerase (Figure 1) [14,16]. Phylum *Lenarviricota* includes ssRNA with positive (+) genome polarity, such as RNA bacteriophages from family *Leviviridae* and their eukaryote-infecting descendants, *Narnaviridae* and *Mitoviridae*, simple RNA viruses without a capsid that are transmitted vertically. Phylum *Pisuviricota* is sometimes called Picornavirus supergroup, and it includes five orders of (+) ssRNA viruses and one order of dsRNA viruses with bipartite genomes, *Durnavirales*. Phylum *Kitrinoviricota* includes only (+) ssRNA viruses infecting eukaryotes, mostly terrestrial plants, and animals. Phylum *Duplornaviricota,* the fourth branch, consists of dsRNA viruses infecting both eukaryotes and prokaryotes. Finally, the fifth branch forms phylum *Negarnaviricota* and includes all known RNA viruses with negative (−) genome polarity [14,16].

RNA viruses in the aquatic environments may infect a wide range of hosts with important consequences. RNA viral pathogens of marine mammals, seabirds, fish, crustaceans, and bivalves can decimate the populations of wild and farmed marine animals with large economic and ecological impacts [17]. RNA viruses have important ecological roles in regulating the structure of protist plankton communities in the oceans. Viral infection of diatoms and copepods, which are mostly infected with RNA viruses, can impact carbon remineralization via the “virus shunt” and carbon export via the “viral shuttle”, thereby altering the efficiency of the biological carbon pump (Figure 2) [1,4,18,19]. The analysis of eukaryotic virus composition showed that eukaryotic viral composition can predict carbon sequestration [18], possibly by affecting the flux through the viral shunt and viral shuttle. Yet, despite their fundamental roles and the enormous diversity of protistan species, only 12 RNA viruses and 41 DNA viruses [6,7,20] have been isolated from protistan hosts due to cultivation challenges.

Routine methods for assessing viral abundance in aquatic environments, such as epifluorescence microscopy and flow cytometry, currently lack the sensitivity to detect the small genomes of RNA viruses [21,22]. Consequently, the abundance of RNA viruses in the virioplankton is presently unknown. Yet, there is some evidence that RNA viruses may exceed DNA viruses in abundance at times [23,24]. Marine RNA viruses are small in size, have a lytic cycle, and have very high burst sizes comparing to the DNA viruses, releasing 1000 to 10,000 particles per infected cell, which could affect their abundances [17,25]. Taken together, these results may indicate that abundances and ecological significance of RNA viruses may be underestimated in aquatic ecosystems.

Traditionally, RNA metaviromics has been the key method to study diversity and dynamics of aquatic RNA virus communities. A comprehensive review summarized gene-marker studies targeting the RNA-dependent RNA polymerase gene and the first RNA metaviromics studies revealing consistent dominance of picorna-like RNA viruses in the marine environments [25]. Knowledge about the protist-infecting RNA viruses isolated through culturing was recently summarized by Sadeghi et al. [20]. Metatranscriptome sequencing greatly expanded known RNA viral diversity associated with invertebrate holobionts, bacteria, plants, and eukaryotes in the soil [9,26,27,28,29]. More recently, a growing number of studies have successfully explored the potential of metatranscriptome mining for RNA virus detection and discovery in aquatic ecosystems. To our knowledge, no reviews have systematically documented this progress from a metatranscriptomic point of view. Here, we describe the most commonly used meta-omic methodological approaches employed in aquatic RNA virology and examine their strengths and limitations. Throughout this review, we refer to meta-omic approaches as viromics, metatranscriptomics, and viral dsRNA sequencing.

We explore in detail 15 recent studies conducted in aquatic environments and 4 studies focusing on individual protists and macroalge, in culture and in natural environments, and discuss how they have broadened our understanding of RNA viral diversity. The focus of this review is on RNA viruses, but if other viral groups were concurrently detected in a referenced study, they are discussed as well to get a more comprehensive overview of the viruses present in a specific environment. Finally, we address the remaining challenges and provide suggestions for future explorations of the ecological relevance of RNA viruses in aquatic ecosystems.

## 2. Methodological Challenges in the Study of Aquatic RNA Viral Communities

Two commonly used approaches for studying viral diversity in aquatic environments are metaviromics (i.e., sequencing of enriched virus particles) and data mining of environmental metagenomes and metatranscriptomes for virus discovery (Figure 3). These approaches are so successful that uncultivated viral genomes of DNA viruses now greatly outnumber cultivated viral isolates in the public databases [30].

Metaviromics (or viral metagenomics) is the method of sequencing extracellular total virus-like particles of either DNA or RNA viruses (Table 1; Figure 3). Enrichment of extracellular, free-floating virus-like particles is achieved through four steps: (i) removal of host cells and their nucleic acids by filtration through 0.22 µm filters, (ii) concentration of the viral fraction via tangential flow filtration [31] or iron flocculation [32], (iii) gradient centrifugation and/or nuclease treatment to eliminate contaminating non-encapsidated (host) nucleic acids [33], and (iv) nucleic acid extraction from virus-like particles [34]. Due to technical challenges, such as filtering large volumes of water and low nucleic acid yields after the extraction, studies using this approach typically target either DNA or RNA viruses. An advantage of the viral enrichment process is an increased proportion of viral reads resulting in improved genome assemblies that enable sensitive detection even of low-abundance viruses [33,34]. Limitations of this method include cellular contamination of the viral metagenomes, which can hinder the data analysis [35], and the fact that the filtration process will systematically remove viruses larger than 0.22 µm, such as giant viruses or long filamentous viruses such as archaeal *Lipothrixviridae* or bacterial *Inoviridae* that can be up to 3.5 µm long [13,30,36]. The latter technical limitation should not affect extracellular RNA viruses since the particle size is consistently around 30 nm, but the removal of the cellular fraction will exclude RNA viruses that do not possess a capsid, do not go through an extracellular state, and are vertically transmitted [37,38,39]. Conventional extraction and sequencing library preparation methods may not suit all genome types, limiting the diversity of viruses that can be detected. For example, ssDNA viruses require a special extraction method and sequencing approach [10] and are therefore systematically omitted from any study targeting dsDNA viruses, and dsRNA viruses may be underrepresented in the RNA viromes due to inefficient conversion to cDNA during the preparation of the sequencing library [40].

Cells of single-celled eukaryotes and prokaryotes from the environment are typically collected onto filters of different pore size, with 0.22 µm filter being the smallest filter size used. Typically, extracellular DNA and RNA viruses will pass through these filters, though aggregates or particle-associated extracellular viruses can still be retained. Viral sequences can be retrieved from both metagenomes and metatranscriptomes derived from the cellular fraction [18,41,42,43,44,45,46,47,48,49,50,51]. 

Metatranscriptomics, which is the sequencing of RNA transcripts from environmental microbial communities, is well suited to study the active viral infection in diverse viral groups with structurally different genomes (Table 1; Figure 3). As a part of the infection cycle, all seven Baltimore classes of viruses produce mRNA intermediate inside infected cells and can be simultaneously captured by metatranscriptomics [52,53]. The additional benefit of this approach is detection of RNA viruses that lack capsid and extracellular stages and are therefore absent from the viral particle fraction, such as *Narnaviridae*, *Hypoviridae,* and *Endornaviridae* [37,38,39]. The main preparations steps include (i) RNA extraction from cells collected on a filter, (ii) DNase treatment to remove the contaminating cellular DNA, and (iii) cellular mRNA enrichment by depletion of highly abundant ribosomal RNAs or by selection of polyadenylated RNA before the library preparation and sequencing. Ribosomal RNA is typically captured by complimentary probes bound to magnetic beads, while polyadenylated RNA is typically selected by binding to oligo-dT probes [52,53]. The removal of ribosomal RNA is a less selective process because it retains all nonpolyadenylated and adenylated cellular RNA. In the context of viruses, this means that with rRNA depletion, RNA viruses that lack a poly-A tail and fragmented or degraded viral RNA genomes can be included in the sample, while during poly-A selection, only RNA viruses (and cellular mRNA) with intact poly-A tail will be retained. This is supported by a recent study demonstrating that rRNA-depleted libraries produced more viral reads, longer contigs, and greater viral richness compared to poly-A-selected libraries [47]. While these results will require additional experimental confirmation as different filter sizes were used for the two treatments, potentially biasing the data, similar results were observed in metatranscriptomes of marine invertebrates [54]. Limitations of this method include a high abundance of rRNA reads in the metatranscriptomes of cellular fractions even after the enrichment steps [55,56] and the fact that the majority of non-rRNA sequences in the cellular metatranscriptomes are of host origin, while viral reads typically constitute less than 1% of the sequenced reads (Table 2 and the references there). 

Additional complexity in metatranscriptomics arises from viral genes being transcribed sequentially and in mono- or polycistronic transcription units, which can lead to uneven genome coverage [52]. The high background of nonviral sequences, combined with the heterogeneity of viral expression, also leads to more fragmented genome assemblies and is the biggest limitation of this approach [61]. Furthermore, for RNA viruses whose genome serves as mRNA, it is not possible to distinguish the viral genome from viral transcripts in the metatranscriptomic datasets. 

Standard methods for reverse transcription used in RNA sequencing are inefficient in transcribing dsRNA molecules as the presence of complimentary strand will prevent the binding of random primers. Therefore, compared to ssRNA viruses, the dsRNA viral lineages may be underrepresented in both metatranscriptomic and viral RNA metagenomic datasets [40]. An approach that might alleviate this problem is dsRNA sequencing (Table 1; Figure 3). Presence of long dsRNA in cells is a unique hallmark of viral infection. Double-stranded viral RNA in cells originates from genomes of dsRNA viruses or from replicative intermediates of ssRNA viruses, with the exception of retroviruses. For DNA viruses, bidirectional transcription (sense and anti-sense direction) of overlapping regions can lead to complimentary transcripts that will form a dsRNA duplex. However, only dsRNA and (+) ssRNA viruses produce dsRNA in amounts detectable by immunofluorescence [62]. Separation of viral dsRNA from total RNA via cellulose chromatography or more recently through immune-based capture, followed by dsRNA sequencing, has been widely used for virus detection and identification in plants and fungi and has led to the discovery of many novel viruses [29,63,64]. Immuno-based capture of dsRNA with anti-dsRNA antibodies was used to pull down multiple (+) ssRNA viral genomes with high recovery (31–74% viral reads) from singular plant samples, including the near-complete genome of an unknown virus [63].

Recently, a novel dsRNA sequencing approach, fragmented and primer-ligated dsRNA sequencing (FLDS), has been developed and applied to marine environmental samples, demonstrating the method’s potential to capture RNA viral diversity that previously went undetected [56]. After conventional total RNA extraction from either viral concentrates or cells, extracted total RNA is enriched for dsRNA via cellulose chromatography. Primers are ligated to the ends of fragmented dsRNA, which is then denatured, and reverse transcription into cDNA with complementary primers is initiated from the ends of the molecules. RNA is degraded with RNase and a complimentary DNA strand synthetized by PCR. Using the FLDS method, on average, 11.3–36.6% of reads obtained from the cellular fraction may be identified as viral as opposed to 0.1% using the classic RNA sequencing approach with rRNA depletion. Similar trends were observed for the viral fraction, where viral reads comprised 35% of total reads with the FLDS method, and only 5% with classic RNA sequencing. A very low number of reads were assigned to rRNA [56]. Overall, virus enrichment through dsRNA purification might be an interesting avenue to explore for research of aquatic RNA viral communities, followed by either standard dsRNA sequencing used in plant virus detection [64] or the FLDS dsRNA sequencing method [56].

In synthesis, there is currently no method that will comprehensively capture all viral diversity. The comparative study of RNA viromes and metatranscriptomes has highlighted tremendous differences between the fractions as well as between different methodological approaches, which will be further discussed below [56]. This observation is not unique to RNA viruses because, for example, the genomes of uncultivated DNA viruses (>10 kb) in the Tara Oceans dataset recovered from virus-enriched fractions and by metagenome mining exhibit huge differences in diversity with an average of 75% and 90% unique sequences in the microbial and viral fractions, respectively [30]. This illustrates that every detection method comes with its biases and using only one approach may show an incomplete or biased snapshot of natural viral communities, underscoring the importance of using multiple approaches.

## 3. Environmental RNA Metaviromics: Lytic Positive-Sense ssRNA Viruses Dominate Pelagic but Not Benthic RNA Viral Assemblages

The first environmental viral metagenomic surveys consistently showed that most sequences in the RNA viromes were unknown. Among known sequences, typically more than 90% of RNA viral sequences identified as (+) ssRNA viruses from order *Picornavirales* in temperate, tropical, and polar oceans, often clustering with known RNA viruses that infect diatoms. Very few sequences of dsRNA viruses were typically recovered, and RNA phages or negative-sense RNA viruses were absent. Only five RNA metaviromic studies in the water column have been performed to date, predominantly in coastal productive environments, with only one study having an extensive geographical distribution [11,24,58,59,60], Recently, the first RNA metavirome study focusing on deep sea sediments was published as a preprint (Figure 4; Table 2) [57].

In RNA metaviromics study of coastal, temperate waters of British Columbia, 98% of the identifiable reads represented (+) ssRNA viruses [59]. In the Jericho Pier library, viral sequences from the order *Picornavirales* were most abundant, mainly consisting of families *Marnaviridae* and *Dicistroviridae*. Two complete viral genomes (JP-A and JP-B) were phylogenetically similar to the diatom-infecting *Rhizolenia setigera* RNA virus RsRNAV and could have a protist (diatom) host. In the Strait of Georgia library, the majority of viral sequences fell into *Tombusviridae*, a family of viruses from the phylum *Kitrinoviricota* [59]. Tropical seawater viral RNA metagenomes of Kaneohe Bay, Hawaii again confirmed the dominance of the order *Picornavirales*, which encompassed 95% of the assignable reads, with a minority of reads being identified as dsRNA viruses. Similarly, the full genomes recovered from the dataset were phylogenetically related to diatom-infecting viruses, or viruses from families *Marnaviridae* and *Dicistroviridae,* consistent with the previous RNA metaviromic study in temperate waters [60]. 

The trend continued with the overwhelming majority of identifiable RNA viral sequences (97.8%) from polar RNA viral metagenomes collected during a spring diatom bloom in West Antarctica Peninsula [24] being classified as (+) ssRNA from the order *Picornavirales*. Out of five full genomes assembled from the dataset, three (PAL128, PAL156, and PAL473) clustered together with diatom-infecting RNA viral isolates or marine RdRp sequences, and two (PAL 438 and PAL_E4) contained highly divergent RdRps. There was a smaller proportion of viral sequences identified as *Marnaviridae* or *Dicistroviridae* compared to the tropical RNA viral metagenomes of Kaneohe Bay. Instead, the relative abundance of diatom-infecting RNA viruses from the genus *Bacilarnavirus* surpassed 90% in three samples or reached around 50%, with unclassified *Picornavirales* taking up the rest. The relative abundance of RNA viruses in the total virioplankton was estimated to range 2–79% in samples on the basis of nucleic acid content of gradient-purified fractions [24].

The largest spatially extensive study of RNA viral metagenomes in the water column to date explored the biogeography of the six marine picorna-like RNA viruses: three complete genomes of picorna-like viruses recovered in this metagenomic study itself [58], two complete picorna-like genomes recovered in a previous metagenomic study [59], and one genome originating from an isolate of HaRNAV, which is genetically more divergent and infects a raphidophyte. Out of the three full genomes recovered in this study, two of them have the highest similarity to PAL 156 genome and one has the highest similarity to PAL 128, the full-length genomes recovered in the temporal study of Antarctic diatom bloom [24] that are thought to be infecting diatoms. Reads from global metagenomic datasets comprising marine, freshwater, and reclaimed water RNA viromes were recruited to these genomes, revealing high geographical and temporal variation of RNA viruses. On a temporal scale, samples collected within a year at the same sample site, Jericho Pier in British Columbia, differ remarkably in distribution of six genomes. In 2013, raphidophyte-infecting HaRNAV was the most abundant virus at Jericho Pier, and in 2014, it was diatom-infecting BC-2 and BC-3 viruses, suggesting that the temporal distribution of RNA viruses in one location is controlled by biotic factors such as host availability. On a spatial scale, the diatom-infecting JP-A was the most abundant genome type off the coast of South Africa, and the raphidophyte-infecting HaRNAV genome dominated the samples from Bering Sea, Laguna Madre in the USA, and samples from Peru and Chile. In the samples from Queen Charlotte Strait and Johnstone Strait in Canada, reads were more equally distributed between diatom-infecting JB-A and JP-B and raphidophyte-infecting HaRNAV genomes. Plenty of viral sequences mapped to the six genomes with low amino acid identity, demonstrating ubiquity and richness of aquatic picorna-like viral communities worldwide. Even if the study did not explore the full diversity of RNA viruses at each sampling location, but instead focused on six selected picorna-like viruses, it demonstrated how dramatically the abundance of certain RNA viral groups can vary between locations [58]. 

More recently, a study recovered 4593 near-full-length RdRp sequences clustered within 2192 clusters at 75% amino acid identity (AAI), in one water sample from a single location in China [11]. Consistent with previous studies, no (+) ssRNA enveloped viruses or (−) ssRNA viruses have been detected and only six RNA viruses were identified as dsRNA viruses, supporting the idea that positive-sense RNA viruses are dominant in the marine environment. In contrast to previous studies, all three phyla of (+) ssRNA viruses were present in the samples, with the order *Picornavirales* constituting only around 26% of RdRp sequences in the sample. Within the phylum *Pisuviricota*, the largest number of RdRp sequences (*n* = 854) grouped with the aquatic picorna-like viruses (order *Picornavirales*) encompassing *Marnaviridae* and other protist-infecting viruses. *Protopotyviruses*, ancestors of complex plant-infecting *Potyviruses*, formed a new clade that was not previously detected in marine environments. Within the phylum *Kitrinoviricota*, many highly divergent RdRp sequences formed a new, phylogenetically well supported clade with previously “orphan” RdRPs of RNA viruses found in marine invertebrates and wastewater, with multiple new orders and families such as *Weiviruses*, *Yanviviruses*, *Zhaoviruses,* and *Shangaiviruses,* significantly expanding the presence of this phylum in aquatic environments, previously limited to detection of *Tombusviridae* in one metagenomic library. Within the phylum *Lenarviricota*, a high number of sequences had similarity to the *Leviviridae*, (+) ssRNA bacteriophages, and the simple eukaryote-infecting *Ourmiaviridae*, documenting for the first time the presence of (+) ssRNA bacteriophages as members of marine RNA viral communities. Capsidless endogenous RNA viruses, such as *Mitoviridae* and *Narnaviridae*, were absent from the extracellular RNA virome as they would have been removed together with cells during filtration. The largest RdRp cluster was affiliated with the *Marnaviridae*, viruses of diatoms and stramenopiles and, together with the use of the alternative (ciliate) genetic code, might suggest that a considerable fraction of RNA viruses infect protists [11].

The high numerical abundance of viral particles and the tremendous impacts of viral infection on bacterial and archaeal communities in marine sediments have been well documented [65,66]. As with the studies of viruses in the water column, research has been predominantly focusing on dsDNA viromes [67,68,69,70], with reports of ssDNA [71] and RNA viruses [57] significantly lagging. A recent preprint by Zhang et al. [57] is the only available RNA metaviromic study in marine sediments, and with 133 samples and almost 3 billion of raw reads, is the largest marine RNA virome dataset to date. Sediments were collected between 2010 and 2018 in a wide range of benthic deep seafloor environments around the globe: hydrothermal vents, cold seeps, sea mounts, and ocean basins at depths between 1000 and 6000 m. The sequenced RNA viromes were searched for marker genes for both kingdoms *Orthornavirinae* and *Pararnavirinae,* RdRp genes, and RT genes, respectively. Most of the identifiable viral RNA sequences were classified as RT-containing *Retroviridae* and *Metaviridae.* Out of RNA viruses with the RdRp gene, the two most abundant families were surprisingly dsRNA viruses from families *Totiviridae* and *Cystroviridae,* in contrast with water column RNA viromes that are typically dominated with ssRNA viruses. Single-stranded RNA viruses from family *Leviviridae* were the most abundant ssRNA viruses detected, supporting the idea that RNA bacteriophages are important members of marine RNA viral communities in the water column as well as in sediments. However, most of the viral sequences could not be assigned to known viral groups, suggesting that the deep sea might represent a rich reservoir of unexplored viral diversity. Alternatively, this might just reflect the current limitations of the available software programs in assigning and classifying viral sequences from environmental samples [57]. 

It is still unclear what fraction of the large differences in the estimates of RNA viral diversity between different studies and different habitats is real rather than a result of methodological biases. RNA viruses from order *Picornavirales* have very high burst sizes [17], and it has been shown that viruses with higher burst sizes can be overrepresented in enriched samples [30]. Consistent with this, in the virus-enriched samples from a diatom bloom from polar waters, the diversity was highest in samples with low abundance of RNA viruses [24]. Similarly, in the virus-enriched samples from temperate waters, the reads in the two libraries were dominated by a few contigs: in Jericho Pier library, 66% of the library assembled in four contigs, and in Strait of Georgia library, 59% of reads assembled into one viral contig [59]. It is evident that the study by Wolf et al. [11] uncovered a staggering diversity of (+) ssRNA viruses in the aquatic environments, broadly expanding the typical range of viral diversity in the first studies. This study had much deeper sequencing depth, as well as a different library preparation method. In the other pelagic viromic studies [24,58,59,60], RNA was preprocessed by sequence-independent, single-primer amplification (SISPA) prior to sequencing, which can lead to overrepresentation of the dominant sequences [72]. More recently, the characterization of the benthic RNA virome by Zhang et al. [57] also had a greater sequencing depth than that of earlier studies, but the results might have been skewed by biases inherent with the use of a Phi29 multiple displacement amplification (MDA) method during library preparation [73,74]. Whether the detected differences in RNA viral diversity are due to potential amplification bias of SISPA libraries and MDA amplification or due to sequencing depth or some other factor will be clarified as more studies become available and published.

## 4. Environmental (Viral) Metatranscriptomics: Sampling with Size Fractionation Improves Metatranscriptome Resolution and Uncovers the Role of ssRNA Viruses

Metatranscriptomics can capture the active viral infection of diverse viral groups and has been increasingly used in studies of RNA virus community diversity and ecological dynamics in the last few years. Most studies were performed in coastal, productive environments, either on temporal or on spatial resolution (Figure 4) and employing different filtration or rRNA removal approaches. Typically, protist and bacterial communities are collected either as “whole” microbial fraction without prefiltration or separated into size fractions using serial filtration, which may allow for higher resolution (Table 2 and the references there). 

One of the first studies that used environmental marine metatranscriptomes followed active infections of diverse viral groups (dsDNA, ssRNA, dsRNA) in two highly productive sites in the US, Narragansett Bay and Quantuck Bay. Seasonal diatom blooms occur in Narragansett Bay, and harmful brown tide blooms caused by unicellular eukaryotic algae *Aureococcus anophagefferens* take place in Quantuck Bay. Active infections of eukaryotic viruses were tracked during the initiation, peak, and demise of the bloom in Quantuck Bay and in the steady-state, non-bloom conditions in Narraganset Bay using marker gene approach [44]. Contigs were searched for marker genes, including the major capsid protein (MCP) for giant viruses, RdRp for RNA viruses, and the viral replicase for ssDNA viruses. Giant viruses from families *Mimiviridae* and *Phycodnaviridae* were continuously present in both locations, with two distinct dynamics. Some giant viruses were steadily expressed across sampling dates, and others exhibited sharp peaks in abundance, or a “bloom and bust” type of expression. In the non-giant virus community, between 62% and 74% of contigs in both bays were unclassified (+) ssRNA viruses in the order *Picornavirales*, followed by classified families *Picornaviridae*, *Secoviridae,* and *Dicistroviridae* from the same order. Few dsRNA viruses resembling *Totiviridae*, *Partitiviridae,* and *Hypoviridae* were detected. The viral community (excluding giant viruses) in Narraganset Bay was steadily dominated (>90%) by *Picornavirales*. In Quantuck Bay, both ssDNA and unclassified *Picornavirales* were present in the *Aureococcus anophagefferens* bloom samples in equal relative abundances, around 50%. Unclassified *Picornavirales* viruses took over (>90%) when bloom disintegrated, possibly infecting other protists in the bloom succession. Co-occurrence analysis between viruses and host marker genes found that most of the hosts might be phytoplankton or heterotrophic protist, but also fungi [44]. Another temporal study of the RNA viral community of three temperate lakes detected RNA viruses resembling families *Picornaviridae*, *Virgaviridae,* and *Reoviridae* with putative protist or invertebrate hosts. The low number of contigs retrieved might be due to low sequencing depth of 1–5 million reads per library [45].

Multiple metatranscriptome size fractions and a metagenome picoplankton size fraction from the Tara Ocean expedition dataset were screened for eukaryotic viruses using marker gene approach: PolB for giant viruses, RdRp for RNA viruses, and RepB for ssDNA viruses [18]. Interestingly, with 975 sequences, RdRp was the most abundant marker gene detected in the metatranscriptomic dataset, followed by 388 PolB sequences of giant viruses and 299 ssDNA viruses. An additional 3846 PolB marker gene sequences of giant viruses were detected in the metagenome picoplankton size fraction. Giant viruses were identified as *Mimiviridae* (mostly algae-infecting), *Phycodnaviridae,* and *Iridoviridae.* Major RNA viral groups in the dataset were *Picornavirales* as well as *Narnaviridae*, *Tombusviridae*, *Virgaviridae,* and dsRNA *Partitiviridae*. Giant viruses and RNA viruses that were positively correlated with carbon export efficiency were abundant in the oceanic regions with high carbon export efficiency, such as the Indian Ocean and the Mediterranean Sea. Hosts of these giant viruses are predicted to be picoplankton (*Mamiellales*) and haptophytes. Hosts of RNA viruses were either copepods or diatoms from the order *Chaetocerales*, both groups being important contributors to the biological carbon pump [18].

More recently, the active viral community within a *Microcystis*-dominated harmful algal bloom in a temperate Lake Tai in China was characterized using the same viral marker genes approach [46]. As expected, most of viral transcripts (48%) present in the metatranscriptome assemblies consisted of bacteriophages infecting *Microcystis,* and RNA viruses constituted the remaining 42% of viral transcripts. Though very diverse, giant viruses take up only 8% of the transcripts. Surprisingly, less than 1% of detected viral transcripts was identified as ssDNA bacteriophages or dsDNA bacteriophages infecting heterotrophic bacteria, though *Microcystis* bloom is typically accompanied with an abundant heterotrophic bacterial community. Interestingly, this is one of the first metatranscriptome studies reporting substantial diversity and abundance of dsRNA viruses. Of the total RNA viral contigs, 28% were identified as dsRNA, mainly *Partitiviridae* and *Picobirnaviridae,* and 72% were ssRNA viruses of the *Picornavirales* order, largely *Marnaviridae* and *Discistroviridae* [46]. This notable increase of detected dsRNA viruses might result from using rRNA depletion rather than poly-A selection [47].

Both metagenomes and metatranscriptomes of the microbial community collected during a diatom bloom succession at Chile Bay in the West Antarctic Peninsula were mined for viruses [48]. Viral reads in the cellular metagenomes of the low-chlorophyll sample were dominated by dsDNA viruses: 82% of viral reads belonging to *Myoviridae*, 9% to giant viruses infecting eukaryotic phytoplankton from the family *Phycodnaviridae,* and 8% to the filamentous ssDNA bacteriophages from family *Inoviridae*. In high-chlorophyll cellular metagenome samples collected during the bloom, the proportion of giant viruses rose to 93% of the DNA reads and previously dominant *Myoviridae* dropped to 3%. In high-chlorophyll metatranscriptomes collected during the bloom, the proportion of (+) ssRNA viruses from family *Picornaviridae* increased to 38%, mostly composed of PAL E4 and PAL 156 viruses, both lytic viruses, and the latter closely related to diatom-infecting RNA viruses from the genus *Bacilarnavirus* [48]. A similar RNA viral community was detected in RNA viromes (<0.22 µm fraction) from Palmer Station at the West Antarctic Peninsula during a summer diatom bloom in 2016 [24]. Taken together, these metatranscriptomic studies show that during a bloom, viral communities that are typically dominated by prokaryotic viruses can shift to eukaryotic viruses, further supporting the ecological relevance of lytic DNA and RNA viruses in bloom disintegration. A metatranscriptome study of virus–host dynamics of the coast of California [49] provided evidence of RNA and DNA viruses tightly controlling protist communities even in non-bloom conditions. After serial filtration, the small fraction showed enrichment for bacteriophages of abundant bacterial groups, while a large fraction was enriched for viruses infecting large phytoplankton, the majority (74%) being dsDNA viruses. The large fraction (<5 µm) was also enriched in fungi, demonstrating fungal importance in the surface ocean as well as their potential as hosts, as they are known hosts of mostly dsRNA viruses. Metatranscriptomic reads from a 60 h sampling study with a 4 h resolution [49] were mapped to the reference genomes of selected RNA and DNA viruses and their host to resolve the diel dynamics of virus–host pairs. An interesting observation from this study was that dsDNA transcripts of dsDNA viruses and their putative host peaked at the same time, without a temporal lag. On the contrary, ssRNA viruses followed a typical “kill the winner” scenario, with the viral transcripts’ peak lagging after the putative host peak. Multiple peaks of viral transcripts of three selected ssRNA viruses and one dsRNA virus during the two sampling days indicated that multiple “boom and bust” cycles of RNA viruses might be happening within short temporal scales. For example, diatom-infecting *Chaetoceros* sp. RNA virus 02 exhibited only one peak, and heterotrophic labyrinthulid protist-infecting *Aurantiochytrium* single-stranded RNA virus 01 peaked five times during the sampling period. This suggests that RNA viruses are important players in regulating the microbial host communities on hourly scales in steady-state systems not perturbed with phytoplankton blooms [49].

A spatial metatranscriptome study in the Baltic Sea at 11 locations spanning a salinity gradient in the brackish Baltic Sea and one freshwater lake identified a wide distribution of ssDNA and RNA populations and high transcriptional activity of fish viruses in different microbial size fractions [50]. Both ssRNA and dsRNA viruses infecting marine protists were detected throughout the dataset, as well as negative-sense RNA viruses from *Mononegavirales* and a high proportion of *Retroviridae*-like sequences. At two high-salinity sites, DNA viruses infecting the protist *Ostreococcus* outnumbered the bacteriophages, amounting to 20%–40% of total viral transcripts, which indicated an ongoing infection. The sample collected in the lake Tornetrask contained virus transcripts mostly belonging to ssDNA bacteriophages from family *Microviridae*, and two samples had a large proportion of ssDNA viruses infecting pigs that could have originated from nearby animal farms. Collectively, these results provide a valuable snapshot into virus diversity at multiple locations and demonstrate that low-abundance viruses can rise to high abundance if conditions allow [50]. 

Studies using the size-fractionated approach show differential enrichment of viral groups in different size fractions [49,50,51]. In the Baltic Sea spatial metatranscriptome dataset and the Tara Ocean Expedition spatial metatranscriptome dataset, RNA viruses were more abundant in libraries originating from the larger cell fractions and are thought to be infecting large unicellular eukaryotes [50,51]. Giant viruses (including *Phycodnaviridae*) and bacteriophages dominated the viral portion of the Baltic Sea metatranscriptome (50–95% viral transcripts) [50]. Similar trends of large DNA viruses dominating viral portions of cellular metatranscriptomes were observed in the Tara Oceans expedition, with 86% of viral genes present originating from giant viruses [51]. In conclusion, these results suggest that size fractionation may improve the resolution and capture a greater diversity of less-abundant viruses. However, care should be taken to avoid using a defined size fraction without a clear understanding of the intended viral targets, because this might lead to more biased results compared to unfractionated samples. 

## 5. Environmental dsRNA Sequencing: The Enrichment of dsRNA from Marine Samples Greatly Expands the Diversity of dsRNA Viruses

The earlier metatranscriptome studies described above recovered mostly ssRNA viruses and very few dsRNA viruses, in agreement with the diversity found in extracellular RNA viromes. A combined metatranscriptomics and RNA metaviromics study compared RNA viruses in the cellular and purified viral particle fractions in four pelagic sampling sites and one coastal sampling site off the coast of Japan [56]. For both cellular and viral fractions, the total RNA was separated by chromatography in ssRNA and dsRNA fractions. The double-stranded RNA fraction contains dsRNA genomes and double-stranded replicative intermediates of ssRNA genomes, and it is enriched for viruses compared to the rRNA-depleted ssRNA fraction that contains mostly cellular mRNA and in which viral ssRNA genomes or transcripts are in minority. Single-stranded RNA and dsRNA were sequenced separately, the latter using the newly developed method, fragmented and primer-ligated dsRNA sequencing (FLDS), resulting in four datasets for each sampling station: cellular ssRNA metatranscriptome, cellular dsRNA metatranscriptome, ssRNA virome, and dsRNA virome. A total of 1270 RNA viral contigs were recovered in the study, mostly originating from dsRNA-enriched metatranscriptomes and dsRNA-enriched viromes (>1000), underlining the efficiency of the dsRNA enrichment procedure compared with the traditional sequencing approach for obtaining viral reads. Due to low number of reads, ssRNA metatranscriptomes were excluded from the analysis. Viral families detected in dsRNA metatranscriptomes were predominantly endogenous ssRNA viruses from family *Narnaviridae*, and dsRNA viruses from families *Picobirnaviridae* and *Reoviridae*. The ssRNA extracellular virome of the coastal site was dominated by (+) ssRNA viruses from the order *Picornavirales,* as reported by other studies [24,58,59,60]. The other sampling sites had too few reads in this fraction for a meaningful comparison. Parallel analysis of the dsRNA virome from the same coastal sampling site revealed an almost equal number of reads belonging to *Picobirnaviridae* and *Reoviridae*, dsRNA viruses with a capsid and an extracellular phase. Many contigs resemble viruses with protist hosts such as diatom colony-associated dsRNA virus-1 (DCADSRV-1) [75] and *Micromonas pussila* reovirus [76]. There are two possible explanations for why FDLS sequencing detects an increased number of dsRNA viruses. The first one is that denaturation of dsRNA before transcription allows for efficient transcription of dsRNA genomes, as they are not efficiently amplified during standard metatranscriptomic library preparation [40]. The second is that there is an amplification bias of the FDLS method, such as that shown for SISPA libraries, which are amplified on a similar principle. Whichever the case, these viruses are present in the aquatic environment and had not been previously detected with standard metatranscriptomic methods. This remains the only study that has explored dsRNA sequencing as a way of studying aquatic RNA viral communities, and it demonstrates the potential of the method to uncover novel RNA viruses and contribute to a clearer picture of the aquatic RNA viral diversity.

## 6. Holobiont Metatranscriptomics: Marine Macroalgae and Cultured Marine Protists Reveal the Broad Distribution of Non-Lytic Strategies in Marine RNA Viruses

There have been several new viral discovery studies in metatranscriptomes of (mostly) axenic protist cultures that have discovered an array of small dsRNA and ssRNA viruses, often with multipartite genomes that coexist with the host and do not cause lysis of the host cell. These persistent infections can be chronic or latent (silent). The advantage of viral discovery in cultured metatranscriptomes over broad environmental metatranscriptomics surveys is that the host can be assigned with certainty and low-titre viruses, which are frequently missed by environmental metaviromics and metatranscriptomics, can be picked up. 

The first report of RNA viruses in the cultured metatranscriptomes came from six species of phytoplankton belonging to two distant phylogenetic groups, green algae (*Chlorophyta*) and chlorarachniophytes [55]. They recovered 18 novel RNA viruses with very low AAI (27–38%). Most viruses were associated with the green algae *Ostrebium *sp., comprising dsRNA viruses falling into families *Partitiviridae* and (+) ssRNA *Mitoviridae*—non-encapsidated small RNA viruses with tiny genomes. Fewer viruses were detected within chloroarachniophytes, possibly due to the clade being divergent and poorly characterized itself, and the highly divergent RdRps might have gone undetected. Additionally, segmented (+) ssRNA viruses from family *Virgoviridae* were detected along with the first detection of a (−) ssRNA virus resembling family *Bunyaviridae* in phytoplankton culture, though the confirmation that the chloroarachniophyte algae are the true host requires additional work [55]. This came as a surprise because both environmental metatranscriptomes and virus isolates showed evidence that green algae are predominantly infected by giant DNA viruses and 85% of isolated RNA viruses were associated with diatoms [6,55].

Recently, the metatranscriptomes of cultured phytoplankton obtained from the Marine Microbial Eukaryote Transcriptome Sequencing Project (MMETSP) were screened for the presence of viral RNA-dependent RNA polymerase hallmark gene [77]. Thirty new RNA viral species with highly divergent RdRps were observed, on average only sharing 35% AAI with known RNA viruses. RNA viruses were observed for the first time in protist families such as *Haptophyceae* and *Chromeraceae*, *Xantophyceae,* and *Bolidophyceae*, which were previously thought to only host dsDNA viruses. One-third of the newly described viruses belong to (+) ssRNA non-encapsidated viruses from families *Narnaviridae* and *Mitoviridae,* known for persistent lifestyles [37]. The double-stranded RNA viruses discovered were dominated by viruses resembling *Totiviridae* that were previously considered to be fungal pathogens. A member of *Mononegavirales*, a negative-sense RNA virus, was detected in the *Pseudo-Nitzchia* culture, but again, host assignment requires additional experimental work [77].

Two studies screened diatom holobionts and marine macroalge for RNA viruses with the FLDS dsRNA sequencing approach [75,78]. The first one unveiled 22 and the second six full-length RNA viral genomes, encompassing five families of dsRNA viruses (*Totiviridae*, *Picobirnaviridae*, *Cystoviridae*, *Partitiviridae*, and *Endornaviridae*) and four families of ssRNA viruses (*Flaviviridae*, *Narnaviridae*, *Virgaviridae*, and *Hypoviridae*). Typically, the viral sequences were very divergent, exhibiting less than 50% AAI to known viral RdRp sequences [75,78].

These metatranscriptome studies focused on single protist or macroalgae holobionts have demonstrated that (+) ssRNA, dsRNA, and potentially (−) ssRNA viruses are distributed across many diverse protist groups and went previously undetected in “bulk” environmental approaches. These viruses have small genomes (typically <5 kb), may be segmented, and some lack capsid proteins. Viral families *Amalgaviridae*, *Chrysovirdae*, *Totiviridae, Partitiviridae, Endornaviridae, Mitoviridae,* and *Narnaviridae* are all associated with persistent, asymptomatic infections in plants and fungi [29] and likely cause nonlytic, but persistent chronic infection in their marine hosts. 

## 7. Recommendations for Future Studies

Several studies discussed in this review demonstrate the plethora of information available when different sequencing approaches are combined [48,50,56], but these studies are still very rare. It is of utmost importance to conduct more studies that integrate metagenomics, metatranscriptomics, and DNA/RNA metaviromics to fully understand the spectrum of the viral diversity and capture the ecological interactions between viruses and their host. Targeted enrichment of dsRNA viral nucleic acid followed by dsRNA sequencing and addition of long-read sequencing could also complement RNA diversity studies by capturing unknown RNA viruses and alleviate the biases of the current methods. To our knowledge, only one study compared RNA viruses in cellular and extracellular viral fractions simultaneously, and it added a novel perspective by demonstrating that both fractions contain a substantial number of dsRNA viruses [56]. A valuable addition to the molecular toolshed for the study of aquatic RNA viral communities could be the use of nanopore sequencing (Oxford Nanopore Technology, Oxford, UK). This technology can be used to sequence native DNA and RNA molecules as they pass through a nanopore; the nucleotide sequence is determined as a change in the current and is specific to the oligomer passing through the pore. Read length is limited only by fragment length, and with the right sample preparation, reads can reach lengths of megabases [79]. It was shown that the inclusion of long reads benefits contig length and captures more DNA viral diversity [80,81,82]. Incorporating long reads in the studies of RNA viral diversity may improve sequence assemblies and/or enable recovery of full-length RNA genomes from environmental samples in a single long read and therefore circumvent the need for assembly. This may help to identify novel viruses and strains. Sequencing of native RNA molecules could also help to overcome biases associated with reverse transcription and cDNA amplification [83]. The constraints of the long-read approach are the amount and quality of starting RNA material.

Despite the immense new RNA viral diversity uncovered with multi-omic approaches, it is evident that RNA viruses in aquatic environments are very sparsely sampled (Table 2; Figure 4). In addition to conducting diversity surveys, more ecology-focused studies are needed to understand the ecological patterns of RNA viruses in aquatic environments. There was only one RNA metaviromics temporal study conducted during Antarctic diatom bloom [24] and one larger spatial study [58], with no continuous temporal studies on a weekly, monthly, or yearly timescale. Metatranscriptomic datasets are more temporally oriented, typically following a bloom over a few weeks on a weekly sampling basis (Table 2). Short time-series, such as that employed by Kolody et al. [49], demonstrate that RNA viruses can be highly active and dynamic and have diurnal fluctuations, emphasizing the need for more higher-resolution studies. 

Most metatranscriptome studies focus on productive areas during algal blooms (Table 2). Efforts to compare across multiple contexts (bloom vs. non-bloom) and in different biomes will undoubtedly lead to a more holistic and comprehensive understanding of RNA and DNA viral dynamics and impacts. This is a worthy endeavour, as the abundance and activity of RNA viruses, and of viruses in general, are intimately connected to the proper functioning of marine food webs, and future climate change will alter the viral-mediated control on biogeochemical cycling [84], with potentially unpredictable consequences.

Another area that deserves attention is standardization and validation of the pre-sequencing and sequencing sample preparation protocols. Enrichment methods, extraction kits, and library preparation can significantly affect the number of viral reads, and consequently the assembly, and skew the relative abundances of particular viral groups and diversity of the viruses present in the sample [47,85]. To ensure robust comparisons, protocols should be compared and validated with mock viral communities to quantify the efficiency and identify the biases of the methods, as previously done for the human virome [85,86,87,88]. Lastly, both metaviromics and metatranscriptomics depend on identifying homology with viral sequences present in the database, and highly divergent viral sequences with little or no sequence homology cannot be readily identified. Even in enriched samples, most reads have low similarity to known nucleotide or protein sequences. Therefore, novel bioinformatic approaches for detecting highly divergent RNA viruses are required to further extend the boundaries of our knowledge about RNA viral diversity and evolution [53].

## 8. Conclusions

Multi-omic sequencing approaches have advanced our understanding of RNA viral diversity and expanded our understanding of their ecological roles in aquatic ecosystems. Three common messages have emerged from the synthesis of all these studies: (1) lytic ssRNA viruses from the order *Picornavirales* are a stable component of the aquatic RNA viral communities, but are perhaps not as dominant as previously thought. (2) All three phyla of positive-sense ssRNA viruses are represented in comparable proportions in the aquatic ecosystem as well as the dsRNA viruses. Highly divergent RNA viruses are being discovered frequently, and are infecting protist hosts that were previously thought only to host DNA viruses. (3) Negative-sense ssRNA viruses, which are traditionally associated with multicellular animals and plants, were found for the first time in the metatranscriptomes of some single-celled eukaryotic phytoplankton cultures. The articles discussed above demonstrate significant progress in characterizing aquatic RNA viral communities, but also highlight how undersampled RNA viral communities are. However, as more data accumulate, we expect to paint a more comprehensive picture of RNA viral diversity, abundance, and host range across the aquatic environments.

## Figures and Tables

**Figure 1 viruses-14-00702-f001:**
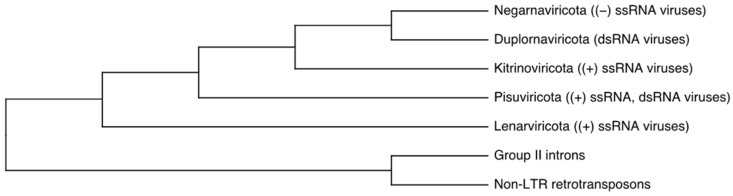
Phylogenetic analysis of RNA-dependent RNA polymerase (RdRp) of RNA viruses with reverse transcriptase (RT) used as an outgroup to root the tree. Five major branches have been assigned a phylum rank by Internationational Committee on Taxonomy of Viruses (ICTV): Branch 1 *= Lenarviricota,* Branch 2 = *Pisuviricota,* Branch 3 = *Kitrinoviricota,* Branch 4 = *Duplornaviricota,* and Branch 5 = *Negarnaviricota.* Adapted from [16].

**Figure 2 viruses-14-00702-f002:**
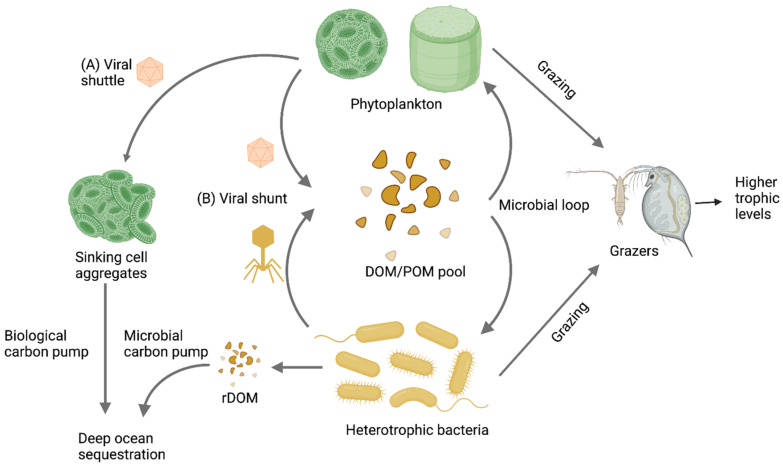
Marine viruses simultaneously control two processes in the carbon cycle: (**A**) the viral shuttle and (**B**) the viral shunt. In the viral shuttle, viral lysis of phytoplankton cells produces sticky aggregates with negative buoyancy that enhance the biological carbon pump by sequestrating carbon in the deep ocean. In the viral shunt, viral lysis of cells has the opposite effect—it diverts the organic matter into a dissolved organic matter (DOM) pool that is rapidly and continuously recycled in the surface waters, preventing its sequestration or uptake by higher trophic levels. Adapted from [4]. Created in BioRender.com.

**Figure 3 viruses-14-00702-f003:**
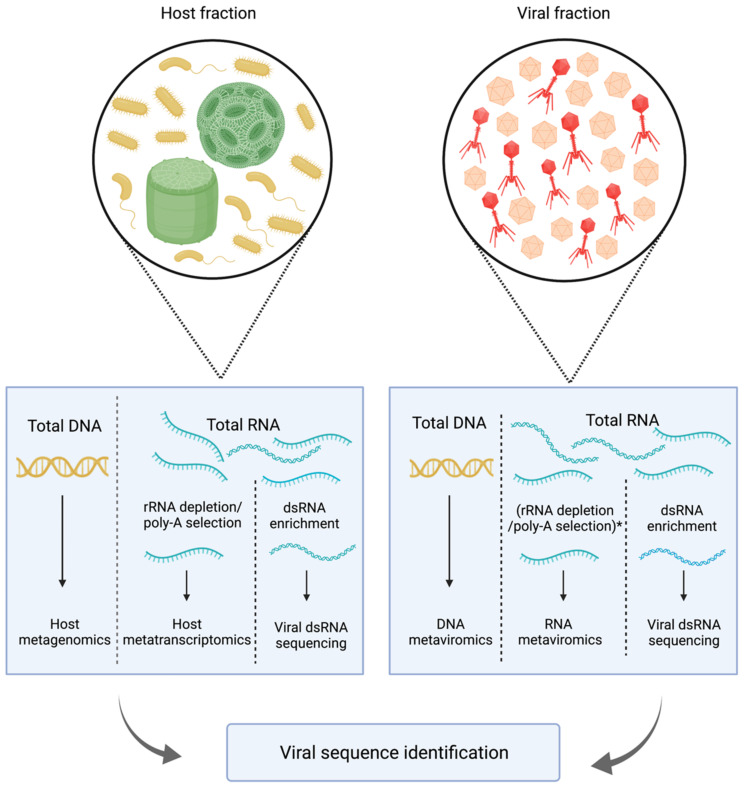
Most commonly used meta-omic sequencing approaches in marine virology. Host fraction (>0.22 µm) metagenomes and metatranscriptomes are mined for viruses, or DNA or RNA is extracted from the enriched viral fraction (<0.22 µm). * rRNA depletion or poly-A selection for RNA in the viral fraction is typically not performed due to low yields. An alternative approach that can enrich for replicative forms of RNA viruses inside the cells or for extracellular dsRNA viruses is dsRNA sequencing. Adapted from [30]. Created with BioRender.com.

**Figure 4 viruses-14-00702-f004:**
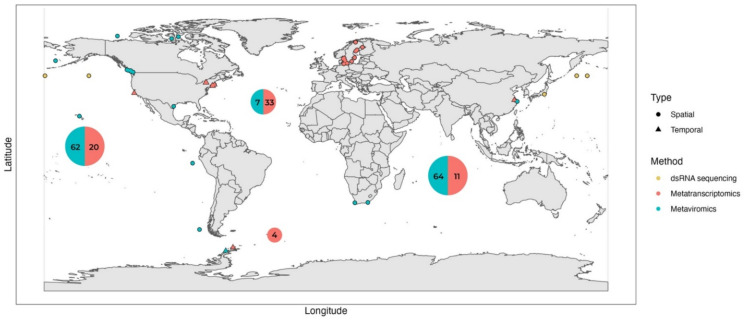
Geographic locations of aquatic metaviromic (blue), metatranscriptomic (coral), and dsRNA sequencing (yellow) studies focusing on RNA viruses reviewed in this paper. Spatially focused studies are represented with circles, and temporally focused studies are represented with triangles. Two global surveys with extensive geographic coverage, Dayang No. 1 cruises and Tara Oceans Expedition [51,57], are represented by large bubbles in each of the five oceans. The size of the bubble is proportional to the total number of samples collected from each ocean, with numbers of benthic virome samples from the Dayang No. 1 cruises [57] indicated in blue, and number of pelagic metatranscriptome samples from the Tara Oceans Expedition [51] in coral. Samples from the Mediterranean Sea were included in the counts of the Atlantic Ocean.

**Table 1 viruses-14-00702-t001:** Advantages and shortcomings of the meta-omic sequencing approaches used in marine viral ecology studies. Adapted from [28].

	Method	Virus Nucleic Acid Detected	Shortcomings	Advantages
	Metaviromics or viral metagenomics	RNA or DNA viral genomes in the extracellular stage	RNA viruses targeted separately than DNA viruses Needs special enrichment for dsRNA and ssDNA Large DNA viruses are filtered out High-burst-size viruses can be overrepresented	Enriched for viral sequences, better assembly
	Metatranscriptomics	Transcripts of (+) and (−) ssRNA, dsRNA, ssDNA, dsDNA	High background of nonviral sequences Potentially fragmented assemblies Misses low-titre viruses Does not distinguish between (+) ssRNA viral genome and transcripts	Captures all types of DNA and RNA viruses simultaneously Captures active infection (for DNA viruses) Can capture RNA viruses without capsids
	dsRNA sequencing	ssRNA as replicative intermediate dsRNA genomes	Not as effective for (−) ssRNA and DNA viruses Cellular metatranscriptomes are removed in the enrichment process	Enriched for RNA viruses Can be used for detection of both extracellular (<0.22 µm fractions) and intracellular RNA viruses

**Table 2 viruses-14-00702-t002:** Basic geographic, ecosystem, and sampling information about the aquatic metaviromic (turquoise) studies, metatranscriptomic (coral) studies, and dsRNA sequencing (yellow) reviewed in this paper. Asterisk * denotes a temporal study in multiple locations.

	Region	Sampling Location	Site Characteristics	Sampling Scheme	Temporal/Spatial	Host or Viral Fractions Collected(Host-Fractionated/Unfractionated)	Relative Abundance of Viral Reads in the Metatranscriptome	Reference
	Global	Multiple locations	Deep sea	133 locations	Spatial	Only viral fraction (RNA)	-	[57]
Polar	West Antarctic Peninsula	Highly productive coastal area	1 location	Temporal	Only viral fraction (RNA)	-	[24]
Polar and temperate	Multiple locations	Multiple site characteristics	11 locations	Spatial	Only viral fraction (RNA)	-	[58]
Temperate	Jericho Pier, Georgia Strait, Canada	Highly productive coastal area	2 locations	Spatial	Only viral fraction (RNA)	-	[59]
Temperate	Yangshan Harbour, Shanghai, China	Brackish coastal area	1 location	Spatial	Only viral fraction (RNA and DNA)	-	[11]
Subtropical	Kane’ohe Bay, Hawai’i	Coastal area	1 location	Spatial	Only viral fraction (RNA)	-	[60]
	Subpolar	Honshu, JapanJamstec cruise	Coastal and pelagic	5 locations	Spatial	Host: Unfractionated>0.22 µmdsRNA + ssRNA metatranscriptomes Viral fraction (RNA) dsRNA + ssRNA viromes	0.1% of ssRNA metatranscriptomes1.3–36.6% of dsRNA metatranscriptomes	[56]


	Polar	Chile Bay, Antarctica	Highly productive coastal area	1 location2 samples, 3 weeks apart	Temporal	Host: Fractionated8 µm–0.22 µm fraction	0.04–0.05%(rRNA depletion)	[48]
Temperate	Baltic seaLake Torneträsk	Eutrophic, mostly brackish	11 location (2 depths)	Spatial	Host: Fractionated200–3.0 µm3.0–0.8 µm0.8–0.1 µm+ viral fraction (DNA)	3.2%(separate rRNA depletion and poly-A selection libraries)	[50]
Temperate	Narragansett Bay (NB)Quantuck Bay (QB), USA	Eutrophic (bloom) coastal	2 locationsNB - 5 samples during 4 weeksQB -3 samples within a week	Temporal *	Host: FractionatedQB5–0.22 µmNB>5 µm	0.043–2.4% (poly-A selection)	[44]
Temperate	Lake Tai, China	Hypereutrophic (bloom) lake	9 locations1 × monthly for 5 months	Temporal *	Host: Unfractionated>0.22 µm	0.02%(rRNA depletion)	[46]
Temperate	Owasco Lake Seneca Lake Cayuga Lake, USA	Mezotrophic to eutrophic lakes	3 locations1 × monthly for 10 months	Temporal *	Host: Fractionated5–0.22 µm	0.6%(rRNA depletion)	[45]
Temperate	California Current, USA	Oligotrophic, with upwelling	1 locationEvery 4 h for 60 h	Temporal	Host: Fractionated>5 µm 5–0.22 µm	Not reported	[49]
Temperate	Quantuck Bay (QB)Tiana Beach (TB), USA	Eutrophic (bloom) coastal	2 locationsQB 1× weekly for 10 weeksTB—1× weekly for 8 weeks	Temporal *	Host: Unfractionated>0.22 µm (for rRNA reduction)>1 µm (for poly-A selection)	0.33–0.53% in rRNA depleted libraries0.02–0.023% in poly-A selected libraries	[47]
Global	Multiple locations	Multiple site characteristics	68 locations2 depths	Spatial	Host: Fractionated2000–180 µm180–20 µm20–5 µm5–0.8 µm	0.0006% to 0.4% (poly-A selection)	[18,51]

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
