# Peer review of "RNA Viruses in Aquatic Ecosystems through the Lens of Ecological Genomics and Transcriptomics"

_viruses, 2022, doi:10.3390/v14040702_

Round 1
Reviewer 1 Report
Kolundzija at al. review the current state of knowledge in RNA viromics of aquatic environment. The authors provide a detailed summary of several largest-scale studies and overview the state of the art and the inevitable limitations of the existing methods of analysis. Overall the review is well-written and very informative.
I have only a few minor issues with the manuscript:
l. 42. "viruses infecting Bacteria and Archaea are almost exclusively double-stranded DNA viruses" is an overstatement (RNA and ssDNA bacteriophages were known from very early days and recent environmental studies keep expanding the set); I would suggest using a milder statement,
l. 394. "with 975 RdRp was the most abundant marker gene" - probably misses a word ("with 975 sequences, RdRp was").
l. 396. "Additional 3846 giant viruses were 396 detected". It is unclear if this refers to the number to markers, contigs or virus particles.
Author Response
We thank reviewer 1 for the constructive comments. The following edits were performed.
Line 42. Text from lines 42-45 has been rephrased according to reviewer’s suggestions. Instead of
“Yet, viruses infecting Bacteria and Archaea are almost exclusively double-stranded DNA viruses [6]. In contrast, viruses infecting marine protists have very diverse genomes comprising from double-stranded (ds) or single-stranded (ss) DNA or RNA, and multiple viral types are able to infect the same protist species [7], [8].”
It says
“They infect cellular organisms in all 3 domains of life, as well as other viruses, and a single species can be infected by multiple viruses. In aquatic enviroments, eukaryotic phytoplankton is known to host very diverse viral communities [4-5]. Among aquatic bacteriophages, dsDNA viruses are the most extensively studied and were thought to dominate prokaryotic virosphere [6]. Reports of ssRNA and ssDNA bacteriophages in environmental metaviromic and metatranscriptomic datasets [7–11], challenged this idea and suggested that bacteriophages have equally diverse genome architecture as eukaryotic viruses.”
The edited text was inserted in lines 36-43 for more logical flow of information.
Line 394 Corrected to “with 975 sequences, RdRp was”. With the new edits, the text is now in line 433.
Line 396. "Additional 3846 giant viruses were 396 detected". It is unclear if this refers to the number to markers, contigs or virus particles.
Corrected to:
Line 435-436. Additional 3846 polB marker gene sequences of giant viruses were detected in the metagenome picoplankton size fraction.
Reviewer 2 Report
Review of the manuscript entitled “RNA Viruses in Aquatic Ecosystems through the Lens of Eco-2 logical Genomics and Transcriptomics” by Kolundzija et al., submitted to Viruses to be considered for publication as a Review
I commend the Authors for this very interesting review on aquatic RNA viruses. I found the work complete and well written. As a benthic microbial ecologist, I expected to see also something about sediments and especially the deep sea, where viruses (possibly including RNA viruses) are more abundant (per volume unit) and can influence their hosts. I am referring to several papers that showed the large diversity, activity and large role of benthic viruses, just to cite some:
Danovaro, R., Corinaldesi, C., Rastelli, E., & Anno, A. D. (2015). Towards a better quantitative assessment of the relevance of deep-sea viruses, Bacteria and Archaea in the functioning of the ocean seafloor. Aquatic Microbial Ecology, 75(1), 81-90.
Zheng, X., Liu, W., Dai, X., Zhu, Y., Wang, J., Zhu, Y., ... & Huang, L. (2021). Extraordinary diversity of viruses in deep‐sea sediments as revealed by metagenomics without prior virion separation. Environmental Microbiology, 23(2), 728-743.
Yoshida, M., Mochizuki, T., Urayama, S. I., Yoshida-Takashima, Y., Nishi, S., Hirai, M., ... & Takai, K. (2018). Quantitative viral community DNA analysis reveals the dominance of single-stranded DNA viruses in offshore upper bathyal sediment from Tohoku, Japan. Frontiers in microbiology, 9, 75.
Danovaro, R., Dell’Anno, A., Corinaldesi, C., Rastelli, E., Cavicchioli, R., Krupovic, M., ... & Prangishvili, D. (2016). Virus-mediated archaeal hecatomb in the deep seafloor. Science Advances, 2(10), e1600492.
Li, Z., Pan, D., Wei, G., Pi, W., Zhang, C., Wang, J. H., ... & Dong, X. (2021). Deep sea sediments associated with cold seeps are a subsurface reservoir of viral diversity. The ISME journal, 15(8), 2366-2378.
Bäckström, D., Yutin, N., Jørgensen, S. L., Dharamshi, J., Homa, F., Zaremba-Niedwiedzka, K., ... & Ettema, T. J. (2019). Virus genomes from deep sea sediments expand the ocean megavirome and support independent origins of viral gigantism. MBio, 10(2), e02497-18.
Rastelli, E., Corinaldesi, C., Dell'Anno, A., Tangherlini, M., Martorelli, E., Ingrassia, M., ... & Danovaro, R. (2017). High potential for temperate viruses to drive carbon cycling in chemoautotrophy‐dominated shallow‐water hydrothermal vents. Environmental microbiology, 19(11), 4432-4446.
L639 are we really ready to state “likely” in this context? Available information may still be too scant for such conclusion?
Overall hint, are we ready to hypothesize something regarding the effects of climate change on RNA viruses and related hosts?
Author Response
We thank Reviewer 2 for the constructive feedback and in response to his/her comments we have amended the manuscript as follows:
We have written an additional paragraph about studies targeting deep sea viruses and described a first preprint study describing deep sea diversity of RNA viruses. The study was added to the Table 2 and Figure 4 as well.
To reflect the changes in the manuscript, the subtitle of the RNA viromics paragraph was edited and the new title is: Environmental RNA viromics: Lytic positive-sense ssRNA viruses dominate pelagic but not benthic RNA viral assemblages (line 250)
The following paragraph was inserted in l. 338-359.
The high numerical abundance of viral particles and the tremendous impacts of viral infection on bacterial and archaeal communities in marine sediments have been well documented [60-61]. As with the studies of viruses in the water column, research has been predominantly focusing on dsDNA viromes [62-65] with reports of ssDNA [66] and RNA viruses [44] significantly lagging. A recent preprint by Zhang et al. [44] is the only available RNA metaviromic study in marine sediments and with 133 samples and almost 3 billion of raw reads is the largest marine RNA virome dataset to-date. Sediments were collected between 2010-2018 in a wide range of benthic deep seafloor environments around the globe: hydrothermal vents, cold seeps, sea mounts, and ocean basins at depths between 1000-6000m. The sequenced RNA viromes were searched for marker genes for both kingdoms Orthornavirinae and Pararnavirinae, RdRp genes and RT genes, respectively. Most of the identifiable viral RNA sequences were classified as RT-containing Retroviridae and Metaviridae. Out of RNA viruses with RdRp gene, two most abundant families were surprisingly dsRNA viruses from families Totiviridae and Cystroviridae,in contrast with water column RNA viromes that are typically dominated with ssRNA viruses. Single straned RNA viruses from family Leviviridae were the most abundant ssRNA viruses detected, supporting the idea that RNA bacteriophages are important members of marine RNA viral communities in both water column as well as in sediments. However, the large majority of viral sequences could not be assigned to known viral groups, suggesting that the deep sea might represent a rich reservoir of unexplored viral diversity [44]. Alternatively, this might just reflect the current limitations of the available software programs in assigning and classifying viral sequences from environmental samples.
The closing paragraph was also modified to include the findings of the new study about deep-sea viruses.
Insertion at line 361-363
The stark differences in diversity of RNA viruses between different studies and different environments could be a consequence of natural diversity, but partially caused by inherent methodological biases.
Insertion at line 375-379
More recently, the characterization of the benthic RNA virome by Zhang et al. [44] also had a greater sequencing depth than earlier studies, but the results might have been skewed by biases inherent with the use of a Phi29 multiple displacement amplification (MDA) method during library preparation [68-69].
We have amended Figure 4 and provided a new caption for the figure:
Figure 4. Geographic locations of aquatic metaviromic (blue), metatranscriptomic (coral) and dsRNA sequencing (yellow) studies focusing on RNA viruses reviewed in this paper. Spatially-focused studies are represented with circles, and temporally-focused studies are represented with triangles. Two global surveys with extensive geographic coverage, Dayang No. 1 cruises and Tara Oceans Expedition [44, 52], are represented by large bubbles in each of the five oceans. The size of the bubble is proportional to the total number of samples collected from each ocean, with numbers of benthic samples from the Dayang No. 1 cruises [44] indicated in blue, and number of pelagic samples from the Tara Oceans Expedition [52] in coral. Samples from the Mediterranean Sea were included in the counts of the Atlantic Ocean.
Line 618. The title Methodological recommendations for future studies was changed to Recommendations for future studies
Line 639 are we really ready to state “likely” in this context? Available information may still be too scant for such conclusion?
The reviewer makes a valid point and "likely" was changed to "maybe" (now on line 681)
In response to the comment "Overall hint, are we ready to hypothesize something regarding the effects of climate change on RNA viruses and related hosts?"
We included this text in lines 657-660.
This is a worthy endeavour, as the abundance and activity of RNA viruses, and of viruses in general, is intimately connected to the proper functioning of marine food webs and future climate change will alter the viral-mediated control on biogeochemical cycling [78], with potentially unpredictable consequences